# Research and Application Progress of Laser-Processing Technology in Diamond Micro-Fabrication

**DOI:** 10.3390/mi15040547

**Published:** 2024-04-18

**Authors:** Yangfan Zhang, Shuai Xu, E-Nuo Cui, Ling Yu, Zhan Wang

**Affiliations:** College of Intelligent System Science and Engineering, Shenyang University, Shenyang 110044, China; z3066178033@163.com (Y.Z.); 15940920132@163.com (L.Y.);

**Keywords:** laser-processing technology, diamond, laser cutting, micro-hole forming, micro-groove machining

## Abstract

Laser-processing technology has been widely used in the ultra-precision machining of diamond materials. It has the advantages of high precision and high efficiency, especially in the field of super-hard materials and high-precision parts manufacturing. This paper explains the fundamental principles of diamond laser processing, introduces the interaction mechanisms between various types of lasers and diamond materials, focuses on analyzing the current development status of various modes of laser processing of diamond, briefly discusses the relevant applications in diamond cutting, micro-hole forming, and micro-groove machining, etc., and finally discusses the issues, challenges, and potential future advancements of laser technology in the field of diamond processing at this point.

## 1. Introduction

Diamonds combine many excellent properties that make them one of the most desirable materials in the world. At the same time, diamonds have been called the “most difficult” material to work with. As a third-generation, ultra-wideband, semiconductor material, the diamond has a typical face-centered cubic structure (lattice constant of 0.35668 nm, bond length of 0.155 nm, and bond angle of 109°28′), and is characterized by high hardness (10 on the Moh’s scale), high thermal conductivity (thermal conductivity of 2000 W/mK), and a low coefficient of thermal expansion (coefficient of expansion of 1.2 × 10^−6^ K^−1^). With high transmittance in the UV-to-microwave range (0.225~8000 μm), the diamond is one of the best choices for manufacturing window materials for optoelectronic systems [1] and is widely used in integrated quantum photonics [2], precision optical component sets for high-power lasers [3], microelectromechanical systems (MEMS) [4], and biomedicine and aerospace [5,6,7].

The diamond substrates used for the preparation of optical devices can be divided into two categories, as follows: single crystal diamond (SCD) and polycrystalline diamond (PCD), according to their structures. The substrates are classified as Ⅰa, Ⅰb, IIa and IIb, according to their nitrogen impurity content, with the specific parameters shown in Table 1 [8]. Natural and artificial diamonds are definied according to their source. Artificial diamonds can be produced in large quantities and the material properties can be controlled. Diamond manufacturing methods can be categorized into hydrostatic, dynamic, and low-pressure methods according to the characteristics of the technology used, and into direct, fusion medium and epitaxial methods, according to the formation characteristics of the diamond. High-pressure high-temperature (HPHT) and chemical vapor deposition (CVD) methods are used to make gem-quality diamonds.

At present, the main processing methods for hard and brittle materials are mechanical processing [9], electrochemical processing [10], electric discharge machining [11], ion beam machining [12], chemical etching [13], laser machining [14], and abrasive water jet machining [15]. The high hardness, brittleness, and the corrosion and wear resistance of hard and brittle materials lead to some insurmountable problems that still exist when machining hard and brittle materials with the above-mentioned methods. The cutting of hard, brittle materials using cutting tools causes problems such as tool wear, microcracks and microchipping, and is time-consuming. Machining hard and brittle materials by mechanical grinding or polishing can result in high surface quality, but cumbersome grinding and polishing processes result in long processing times and high processing costs, and grinding produces scratches, micro-cracks, chalking, fine plastic deformation, and significant residual stresses on the surface. Electrochemical machining and EDM require the material to be electrically conductive, and the Joule heat generated by spark machining leads to instability, poor machining accuracy, and high energy loss in the machining process. Ion beam processing can achieve high-precision and high-quality processing of hard and brittle materials, but the ion beam processing efficiency is very low, and the equipment is expensive, and needs to be carried out in a vacuum environment. The environmental requirements are harsh. The use of strong acid or alkali chemical reagents can be prepared on the surface of hard, brittle materials and micro-nanostructures, but the chemical wet-etching efficiency is low, and for single-crystal, hard, brittle materials, there are also different crystal-etching rates, resulting in low processing accuracy. In addition, the chemical reagents are also harmful to people and the environment.

The main types of lasers used in current research on the laser processing of diamond materials are shown in Table 2. Most of these have pulse frequencies between 0.1 Hz and 100 kHz, and the shortest pulse widths are on the femtosecond scale. By choosing the appropriate laser wavelength, pulse width, and power parameters, diamond can be processed with high-quality and special shapes, and its processing accuracy can reach the micrometer or nanometer level. The laser processing of diamond materials has the advantages of flexible processing modes, high processing efficiency, and high precision. It is also a simple, clean process that is environmentally friendly, and is an efficient and controllable method of processing diamonds. This paper reviews the mechanism of diamond laser processing, compares the effects of different types of laser processing, summarizes research on the application of lasers in diamond cutting, drilling, micro-groove molding, etc., and considers the future research direction of diamond laser-processing technology [16].

## 2. Laser and Laser-Processing Mechanism

The diamond sublimation or chemical etching that occurs during laser processing does not directly occur but must first go through the process of a diamond-to-graphite phase transition; this carbon phase transition is one of the key points in the laser processing of diamond. The graphitization behavior of the diamond material reduces processing difficulties [17]. Based on quantum theory, in the interaction between matter and the radiation field, the atoms or molecules constituting the matter can produce excited emissions or the absorption of photons under the excitation of photons, which laid the foundation for the emergence of subsequent lasers. When the frequency of foreign photons meets the requirements of an energy level jump, this will make the atom in the high-energy level of the electron in the excitation of foreign photons to the low-energy level jump, and release and incident photon frequency, propagation direction, phase and polarization are the same as the excited radiation photons [18]. The amplification of the excited radiation, the inversion of the set number, and the critical state of laser oscillation are the three major conditions for laser generation, which corresponds to the three basic components of the laser pump source, the laser working medium, and the resonant cavity [19]. According to the above laser generation principle, the laser is an excited radiation coherent light source with high brightness, high directivity, high monochromaticity and high coherence, and has excellent time and space control performance. The pulse frequency is mostly between 0.1 Hz and 100 kHz, and the shortest pulse width is on the femtosecond scale. In the actual processing of diamonds, by choosing the appropriate laser wavelength, the pulse width and power parameters can used for high-quality and special-shape processing of diamonds, and its processing accuracy can reach the micron level or even the nanometer scale.

### 2.1. Diamond Absorbs Laser Energy

Laser processing is the process of removing surface material by irradiating a solid surface with a laser beam. At low laser fluences, the material absorbs the laser energy and heats up and evaporates. A number of factors affect the interaction between the laser and the solid material, including pulse length, wavelength, laser power, repetition frequency, beam characteristics, and the physical properties of the solid material. Inside the solid, light can occur in transmission, reflection, and absorption. Only the absorbed energy can produce ablation inside the solid, while the reflected and transmitted light affects the shape and position of the preheated volume in the ablation region. As shown in Figure 1A, a single-crystal diamond only has effective absorption of photon energy beyond the energy band gap (5.4 eV) (corresponding to a wavelength of 229 nm) [20]. This indicates that the absorption of laser energy in solid materials is influenced by the material absorption coefficient, and that there is a huge difference between the absorption characteristics of short-pulse and ultrashort-pulse laser irradiation in diamond materials. In practice, 532 nm and 1064 nm lasers are often used for processing, and these wavelengths are not absorbed by impurity-free diamond materials, whose energy band structure is strongly influenced by intrinsic and non-intrinsic defects, including grain boundaries, crystal defects, non-diamond phases, and dopant atoms [21]. These defects also lower the laser ablation threshold of diamonds, with a high concentration of defects corresponding to a low ablation threshold, which favors laser processing.

Multi-photon absorption occurs during the irradiation of ultrashort-pulsed lasers, where electrons can absorb more than one photon and thus be excited, a process that requires high temporal and spatial photon densities in the focused beam [22]. Because the ionization potential of bound electrons is much larger than the energy of a single photon of laser light, the bound electrons are not released under normal circumstances. Multi-photon ionization can be generated when the laser intensity is higher than 10^12^ W/cm^2^, and multi-photon ionization releases bound electrons. Electrons can absorb the energy of several photons at the same time to produce excited electrons, and the coupling of excited electrons and phonons leads to the heating of the crystal lattice, which leads to the occurrence of a phase explosion [23]. Komlenok [24] found that the ablation threshold also varies with pulse width and number of pulses, as shown in Figure 1B. In general, shorter pulse durations always correspond to smaller ablation thresholds. Multiple pulses are usually used in the process to lower the ablation threshold at the ablated surface area. Under prolonged multi-pulse laser irradiation, the cumulative effect of multiple pulses causes the light absorption coefficient of the material to gradually increase until the energy absorbed by a single laser pulse is sufficient to disrupt the lattice bonding of this type of graphitic defect, which results in an ablation effect. At the same time, to make diamond graphitization under low-energy density laser irradiation, it is necessary to increase the number of pulses, and the lower the energy of a single pulse, the greater the number of pulses required [25]. From the above research summary, the future of diamond laser processing is moving in the the direction of short wavelength, narrow-pulse duration processing.

**Figure 1 micromachines-15-00547-f001:**
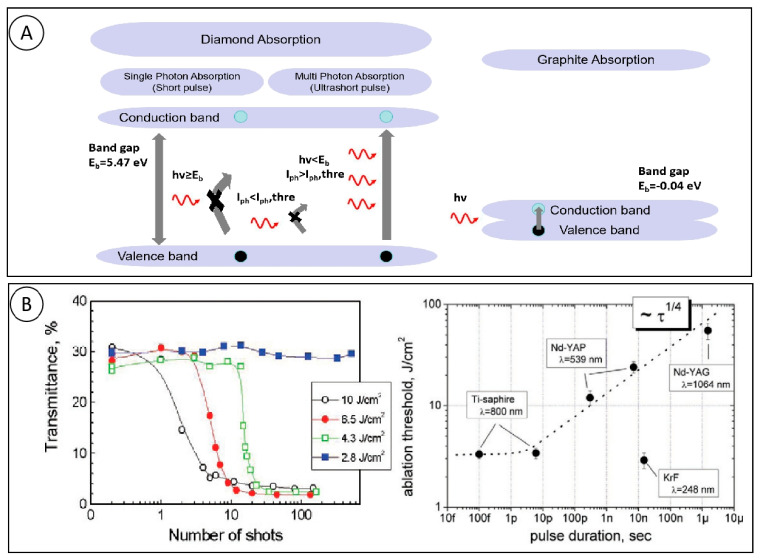
(**A**) Absorption behavior of diamond and graphite; and (**B**) dynamics of CVD diamond transmittance caused by multiple laser irradiations at different fluencies and dependence of CVD diamond vaporization threshold on laser-pulse duration [24].

### 2.2. Changes in the Properties of Photoinduced Diamond

Diamonds are covalent crystals in which one s-orbital and three p-orbitals hybridize to form four sp3 hybridized orbitals of equal energy arranged in an orthotetrahedral configuration, which minimizes the repulsive force on the electrons. Under short-pulse laser irradiation, the diamond temperature rises sharply, making the C–C covalent bond broken and reconnected; three of the four outer valence electrons of each carbon atom 2 s, 2 px, 2 py in the form of sp2 hybridized orbitals in the same plane through the σ-bonding with three carbon atoms form a covalent single bond, with the orbitals between the central axis of the 120° angle. The other 2 pz orbital electrons that are not involved in the hybridization are perpendicular to this plane and form unsaturated π-bonds in the pπ orbitals [26], as the diamond–graphitization transition occurs.

Laser processing of diamond materials is often associated with a reduction in diamond transmittance, which is more pronounced when long wavelength lasers are used. Laser-induced internal damage is the main cause of this situation, and laser light breakdown on the diamond surface can be induced by lasers of different pulse durations. Smedley [27] clearly observed the presence of internal opaque defects in polycrystalline diamond using a picosecond laser with a pulse duration of 266 nm (Figure 2A), and the size and number of laser-induced defects in the polycrystalline diamond were found to be significant at long pulse widths (10 ns) and wavelengths (532 nm). The size and number of laser-induced defects in polycrystalline diamond increase at long pulse widths (10 ns) and wavelengths (532 nm). The 1030 nm near-infrared picosecond laser processing of polycrystalline diamond was investigated by Kononenko [28] and it was concluded that, before the initiation of graphitization and ablation on the surface of the diamond, single or multiple photopenetrations occur in the sub-surface layers of the diamond, and that the effect of each of these penetrations produces micrometer-sized graphitic inclusions. On this basis, they proposed a method of pre-coating an absorber layer (Ti or graphite) on the surface before processing to reduce the subsurface optical breakdown (Figure 2B) [29], and found that the presence of the absorber layer was effective in avoiding the appearance of processing damage in the internal layer of the diamond when laser processing with longer pulse durations (10 ns), while it did not have a significant effect on laser processing with shorter pulse durations. The presence of an absorbing layer was found to be effective in avoiding the appearance of internal diamond-processing damage for longer pulse durations (10 ns), but not for short-pulse durations.

The photoelectric properties of a diamond, including light absorption and the quantum efficiency of photons to electrons, depend strongly on its intrinsic crystal structure and the presence of impurities. Photocurrents in pure diamond crystals without impurities can be induced by photon absorption under UV irradiation, but the presence of impurities and defects within the actual diamond makes it possible to cause photon absorption by visible light as well. This suggests that the presence of defects within the diamond can cause a low-energy light absorption effect, thereby increasing the photoconductivity of the material. Recently, researchers have conducted a number of studies around the use of femtosecond lasers to induce the generation of color centers. V. Kononenko presented the concept of color centers prepared by surface nano-ablation [30,31], which were efficiently and controllably generated by a femtosecond laser acting on the diamond surface, and also explored the influence of the laser parameters on the formation of NV color centers during the laser–diamond interaction, as shown in Figure 2C,D.

### 2.3. Diamond Surface Morphology Change

In diamond laser processing, the shape, intensity, and polarization of the incident laser beam cause differences in the morphology of the processed surface. When the laser is incident at a low-energy density, sp2 graphitization occurs immediately in the heat-affected zone, whereas at higher laser densities, the zone is rapidly sublimated by the pulse width of the incident laser beam [32]. Diamonds produce periodic surface structures when irradiated by a linearly polarized laser. Forster [33] used a femtosecond laser to process a nitrogen-doped Ib-type diamond, and the high spatial-frequency laser induced perpendicular to the direction of the electric-field polarization resulted in the formation of periodic surface structures with a period of 50 nm and 200 nm, in which the 50 nm periodicity was derived from the diamond. The 50 nm periodic structure was derived from the graphitization transition of diamond, while the 200 nm periodic structure was still composed of crystalline diamond.

The periodic structures generated on the surface are much smaller than the incident laser wavelength, and femtosecond lasers produce fewer periodic features than nanosecond and picosecond lasers, which have longer pulse durations, and are therefore more suitable for nanoraster processing. The laser energy density and the number of pulses also have an effect on the resulting nano-periodic structures. Figure 3A shows the surface morphology of a diamond sample irradiated by an 800 nm linearly polarized femtosecond laser at a laser energy density close to the diamond ablation threshold of 1.9 J/cm^2^ and a pulse number of 3000, where a regular 170 nm periodic nanoscale grating exists at the edges of the processed area, and a regular 170 nm periodic nanoscale grating exists at the edge of the processed area at a laser energy density of 2.8 J/cm^2^ and a pulse number of 8000. At a laser energy density of 2.8 J/cm^2^ and a pulse number of 3000, a periodic structure at the edge of the processed region is present with a regular 170 nm periodic nanoscale grating, and at a pulse number of 8000, the periodic structure at the processed region increases to 190 nm [34]. Mastellone et al. [35] used a delayed cross-polarized femtosecond laser pulse train to obtain a micro-nanostructure with two ultrahigh-frequency structural perimeters (Λ ≈ λ/10 ≈ 80 nm) on the surface of the diamond and explored the mechanism of the formation of such a structure, which is illustrated in Figure 3B.

The influence of the main laser process parameters on diamond graphitization is shown in Table 3. The adjustment of the laser process parameters can change the degree of diamond graphitization, so that based on the degree of surface graphitization and the surface morphology of diamond graphitization, the local controlled removal of graphite oxidation is carried out, and the diamond structure left behind forms the functionalized pattern on the surface. The actual laser processing of diamond materials is very complex, and the influence of various environmental parameters is coupled with each other, so it is still very difficult to precisely use the laser beam to process the diamond surface with high precision and controllable functionalized patterns.

The laser processing of diamonds is often accompanied by the appearance of graphitization, and its induction mechanism can be explained from the microscopic point of view, as follows: the huge energy of the laser will stimulate the electrons to undergo an energy level jump, and the electrons will collide with the lattice to warm up and transfer to the lattice so that the lattice will reconstruct, and the phase transition will take place. The laser process parameters have different effects on the graphitization process.

(1)The effect of laser energy density on the thickness of the graphitized layer on the diamond surface is determined by the size of the ablation threshold. Below the ablation threshold, in the laser-irradiated area, there is a potential graphitization only in the internal phase transition. Due to the existence of ablation vaporization, the graphite layer formed on the surface will be further ablated and leave ablation craters. The higher the laser energy density, the more obvious the ablation phenomenon; Due to the existence of ablation vaporization, the graphite layer formed on the surface will be further ablated and leave ablation craters. The higher the laser energy density, the more obvious the ablation phenomenon;(2)The pulse width increases the depth of the heat-affected zone, which increases the ablation threshold, which in turn affects the graphitization rate;(3)The number of pulses also affects the graphitization rate indirectly by influencing the ablation threshold. An increase in the number of pulses leads to a decrease in the ablation threshold and an increase in the diameter of the irradiated zone, thus increasing the thickness of the graphite layer;(4)The laser power and scanning rate together affect the laser energy density, the laser scanning rate decreases will lead to the number of pulses irradiated to a fixed position per unit time increases. Laser energy density, pulse width, number of pulses, laser power and scanning rate are related to each other in the laser processing, which affect the graphitization process of diamond. Laser processing, as a highly efficient method of processing diamond, has received a lot of attention from industry players recently, while still facing a number of challenges.

**Table 3 micromachines-15-00547-t003:** Influence of the main laser processing parameters on diamond graphitization.

Laser Parameters	Effects on Graphitization of Diamond		Related Expressions
$\mathrm{Laser}\;\mathrm{fluence}\;\left(F\right)$	F<$\;\mathrm{ablation}\;\mathrm{threshold}\;(F_{\mathrm{th}}$)	$\mathrm{Thickness}\;\mathrm{of}\;\mathrm{graphite}\;\mathrm{layer}\;\left(d_{g}\right)$ increases with the increase in F	Latent graphitization
	F≥$\;\mathrm{ablation}\;\mathrm{threshold}\;(F_{\mathrm{th}}$)	$d_{g}$ decreases with the increase in F	Macro-graphitization
Laser duration (τ)	τ<10 fs	$d_{g}\;$ $\;\mathrm{is}\;\mathrm{determined}\;\mathrm{by}\;\mathrm{the}\;\mathrm{absorption}\;\mathrm{coefficient}\;\left(\alpha_{g}\right)$ of the graphite-like phase, independent with τ	$d_{g}\approx \frac{0.7}{\alpha_{g}}$
	τ≥10 fs	$d_{g}\;$ is determined by both of τ$\;\mathrm{and}\;\mathrm{thermal}\;\mathrm{diffusion}\;\mathrm{coefficient}\;\;\left(X_{g}\right)$ of the graphite-like phase	$d_{g}\approx 0.7{\left(X_{g}\tau \right)}^{1/2}$
$\mathrm{Pulse}\;\mathrm{number}\;\left(N\right)$	$F_{\mathrm{th}}\;$ decreases with the increase in the effective pulse number N		N<124 $F_{\mathrm{th}}$ drops sharply
			N>486 $F_{\mathrm{th}}$ drops slowly
$\mathrm{Laser}\;\mathrm{power}\;\left(P\right)$	Laser power, laser repetition rate (f $)\;\mathrm{and}\;\mathrm{the}\;\mathrm{beam}\;\mathrm{waist}\;\mathrm{radius}\;\mathrm{of}\;\mathrm{the}\;\mathrm{laser}\;\mathrm{spot}\;(\omega_{0})$ $\;\mathrm{jointly}\;\mathrm{detemines}\;\mathrm{the}\;\mathrm{laser}\;\mathrm{energy}\;\mathrm{density}\;\left(F\right)$		$F=\frac{2P}{f\cdot {\pi \omega }_{0}{\;}^{2}}$
Scanning speed	Pulse number N increases with the decrease in the laser scanning speed		——

## 3. Interaction of Different Laser Types with Diamond Materials

Laser-processing technology, when applied to diamond processing, can produce diamonds with high efficiency high precision. Different laser beam types have very different effects on diamond surfaces. The lasers used for diamond processing can be categorized into “hot processing” and “cold processing” lasers, according to the relationship between the laser pulse length and the size of the atomic lattice collision [36]. The most representative of these are nanosecond and femtosecond lasers. An interaction model of two typical lasers with electrons and lattices is shown in Figure 4. For diamonds, the relaxation time of the electrons and holes is 1.5 ps and 1.4 ps, respectively [37], and heat transfer occurs between the electrons and lattice when the laser reacts with the diamond. For the nanosecond laser, with a longer pulse duration, the laser energy deposited in the electrons is transmitted to the lattice within the time of irradiation of the material by the laser pulse, which leads to the heating of the material and reaches the thermal equilibrium state. There is an obvious thermal effect in the process, so it is called “thermal processing”. For the femtosecond laser, the laser pulse width is less than the time scale of electron–phonon interaction; thus, the laser energy deposited in the electrons cannot be transmitted to the ions in time, when the laser pulse irradiation has ended. At this time, the temperature of the ions is relatively low, so it is called “cold processing”.

The absorption of laser energy by diamonds is a nonlinear absorption process. Usually the energy band gap energy of diamond is larger than the photon energy of the laser; however, under the excitation of a high-intensity laser, the electrons in the material can form free electrons through nonlinear processes such as multiphoton ionization, avalanche ionization and other physical phenomena. At laser intensities below 10^12^ W/cm^2^, free electrons are excited and absorb energy through the reverse bremsstrahlung process, which is repeated, and the increase in the density of free electrons simultaneously leads to avalanche ionization by the absorption of laser energy by the dielectric. The heat conduction formula of a laser acting on the material is as follows [23]:(1)$$\begin{array}{c} \rho c_{\rho }\frac{\partial T}{\partial t}=k\left(\frac{\partial^{2}T}{\partial x^{2}}+\frac{\partial^{2}T}{\partial y^{2}}\right)+\frac{\partial^{2}T}{\partial z^{2}}-\rho c_{\rho }v\left(\frac{\partial T}{\partial x}+\frac{\partial T}{\partial y}+\frac{\partial T}{\partial z}\right)+\omega \left(x,y,z,t\right) \end{array}$$

Equation (1): *ρ* is the density of the material; *t* is the time variable; *k* is the thermal conductivity; *ω* is the heating rate of the heat source per unit time per unit volume of the heat transfer to the material; $c_{\rho }$ is the specific heat capacity; *T* is the temperature; *v* is the laser scanning speed. The conventional heat source for laser energy is a Gaussian-distributed heat source, which can be expressed as follows:(2)$$I\left(r,0\right)=I_{0}exp\left(-2\cdot \frac{r^{2}}{S^{2}}\right)$$

In Equation (2): *r* is the distance from the spot center; *I*_0_ is the spot center power density; and *S* is the spot radius. The Gaussian laser is characterized by uneven energy distribution. Close to the center, the energy distribution is relatively high. On the contrary, the farther away from the center, the lower the energy distribution. Therefore, the Gaussian profile can clearly show the distribution of temperature and pressure, as is shown in Equation (3):(3)$$\begin{array}{c} \omega \left(x,y,z,t\right)=\sqrt{\frac{4ln2}{\pi }}\frac{\left(1-R\right)J}{t_{\rho }\delta }\cdot exp\left[-\frac{x}{\delta }-4ln2\cdot {\left(\frac{t}{t_{\rho }}\right)}^{2}\right] \end{array}$$

In Equation (3): *J* is the laser intensity; *R* is the material surface reflectivity; *t_p_* is the laser pulse width; *δ* is the laser pulse penetration depth. By substituting Equation (3) into Equation (1), the heat transfer equation for Gaussian profile laser processing can be obtained. It can also be concluded that the main effects of laser processing are spot distance, reflectivity, and laser scanning speed.

Assuming a very fast heating process of subsystems of electrons and lattices, the energy conversion process can be described by a one-dimensional diffusion equation for two characteristic temperatures [23]:(4)$$C_{e}\frac{\partial T_{e}}{\partial t}=-\frac{\partial Q_{z}}{\partial z}-\gamma \left(T_{e}-T_{i}\right)+S$$
(5)$$C_{i}\frac{\partial T_{i}}{\partial t}=\gamma \left(T_{e}-T_{i}\right)$$
(6)$$Q_{z}=-k_{e}\frac{\partial T_{e}}{\partial z},S=I\left(t\right)A\alpha exp\left(-\alpha z\right)$$
where *z* is the direction of heat propagation perpendicular to the target surface, *Qz* (*z*) is the heat flux *I*(*t*) is the laser intensity, A=1−R is the surface transmittance, *α* is the absorption coefficient of the material, *C_e_* and *C_i_* are the heat capacities per unit volume of the electrons and ions, respectively, *γ* is the quantity describing the electron–lattice coupling, and *k_e_* is the thermal conductivity of the electrons. *T_e_* and *T_i_* are the temperatures of the electrons and ions, respectively. From the above equations, the heat capacity of the lattice subsystem (phonon component) has been neglected. The electron heat capacity is much smaller than the heat capacity of the lattice; therefore, the electron can be heated to very high temperatures very quickly. When the electron temperature is kept lower than the Fermi temperature, the electron heat capacity and the thermal conductivity of non-equilibrium electrons are expressed as $C_{e}={C^{\prime }}_{e}T_{e}$ (${C\prime }_{e}$ is a constant), $k_{e}=k_{0}\left(T_{i}\right)T_{e}/T_{i}$ [$k_{0}\left(T_{i}\right)$ is the conventional equilibrium thermal conductivity of the metal].

### 3.1. Microsecond Laser Processing

Microsecond lasers have wide pulse widths and are usually suitable for rough machining. Before the advent of mode-locking technology, laser pulses were mostly on the order of microseconds and nanoseconds. The long pulse widths of microsecond lasers are accompanied by strong thermal damage during processing [38,39]. Eberlea [40] characterized the results of microsecond laser processing of polycrystalline diamond composites in comparison with other processing methods, as shown in Figure 5, proving that there is a significant heat exchange in the PCD composites when processed with microsecond lasers. This is different from the nanosecond laser, which has a peak power of a few kilowatts and a long pulse duration, resulting in a deeper heat-affected zone of 6.8 μm.

### 3.2. Nanosecond Laser Processing

At present, nanosecond lasers hold a significant share of the market and are extensively utilized in corporate production due to their advantages of superior stability, low cost, and short processing time [41]. Nozomi Takayama [42] classified the defects produced by nanosecond laser processing of diamond, as shown in Figure 6A, which are categorized into four types (cracking, corrugation, deformation of grooves, and debris deposition), and explained the reasons for the occurrence of each type of defect. Cracks are caused by rapid temperature changes during processing; ripples are formed due to interference of the reflected laser from the groove wall; the deformation of the groove and its deviation from the Gaussian laser is due to enhanced absorption by the laser-induced plasma. There are two main types of debris deposited: rounded graphite-carbon particles and smaller irregular diamond particles. Cadot [43] investigated the Raman spectra of a diamond after processing with different laser fluences, as shown in Figure 6B. At lower laser fluences, there are two characteristic peaks of 1380 cm^−1^ and 1580 cm^−1^ in the Raman spectra, which indicates that the graphite produced by low laser fluences is in the shape of large clusters, and the interference to the crystal structure is small. With the increase in laser fluence, the FWHM variations of the D peak, which represents the disordered graphite structure, and the graphite in the G peak is broadened, which indicates that the number of defects in the lattice increases. Zhen Zhang [44] established a finite element simulation model of a three-dimensional moving nanosecond pulsed Gaussian laser ablation of a single-crystal diamond and obtained the heat conduction and temperature distribution within the single-crystal diamond under different scanning times, as shown in Figure 6C, demonstrating that the model has a good matching relationship with the actual results and that the model has a good prediction ability.

### 3.3. Picosecond Laser Processing

Picosecond laser processing is different from thermal equilibrium ablation with nanosecond lasers and is not exactly the same as cold processing with femtosecond lasers, as the significant reduction of pulse duration greatly reduces the damage caused by the heat-affected zone. Pimenov [45,46,47] used a picosecond laser on the interior of diamond and observed the graphitization phenomenon inside the diamond, as shown in Figure 7A, and based on this, they prepared graphitized microstructural arrays, and investigated the effect of crystal orientation on the ablation process of picosecond laser. At the same time, they found that 3H and NV color centers were generated at the graphitized positions, and the fluorescence of the NV color centers was enhanced by the laser body modification process. Nozomi Takayama [48] used picosecond pulsed laser to process single crystal diamond tools with special structures, and when the laser parameters were: laser fluence 15.3 J/cm^2^ and repetition frequency 100 kHz, the tools could be processed quickly without any edge chipping. Wen Qiu Ling used an infrared picosecond laser to process CVD single crystal diamond microgrooves, and found that only at the edges of the microgrooves there are irregular tiny chipping and microcracks, and at the same time, their simulation results of the temperature field of laser ablated diamond also show that the laser irradiation energy is mainly distributed on the surface of diamond, and the laser energy entering into the interior of the diamond in the form of thermal conduction is very small, which produces the heat-affected zone is also very small. Similar to the nanosecond laser, the interaction between the picosecond laser and diamond is also carried out by surface graphitization, and Raman analysis of the bottom of the processed diamond microgroove, as shown in Figure 7B, shows that the graphite peaks show an obvious red shift with the increase in the picosecond laser energy density [49].

### 3.4. Femtosecond Laser Processing

After decades of rapid scientific and technological development, ultrafast laser technology has created opportunities for the fine processing of diamond materials, and most of the current research on femtosecond laser processing of diamond materials remains at the laboratory stage [50]. In femtosecond laser processing, laser energy is applied to the irradiated area through the photo-induced optical breakdown effect, in which many electrons dissociate, leading to changes in the structure and phase composition, and in the case of diamonds, the transition from the sp3 phase to the sp2 phase occurs, followed by ablation of the material in the irradiated area. For high-quality diamond materials, the absorption of laser energy is nonlinear because the incident photon energy is not sufficient to cause ionization [51], which requires high electric field strength and corresponding laser energy density in the laser pulse. The very small laser spot and the nonlinear effects during femtosecond laser processing limit the processing area [52].

Femtosecond pulsed lasers produce very high power densities (up to several GW) at low average power (100 mW), and are so high that C–C covalent bonds in the diamond lattice are dissociated [53]. The laser-induced diamond–graphitization transition increases the spacing of the carbon atoms, which decreases the density of states and changes the physicochemical properties of the solid. The possibility of heat-affected zone formation is minimized in the very short-pulse duration, and the surface structure of the diamond is precisely processed with minimal thermal damage [54], as shown in Figure 8A. Ogawa [55] compared the material removal rate and surface quality during infrared nanosecond laser and femtosecond laser processing, as shown in Figure 8C, and indicated that the surface quality was higher when femtosecond laser processing was used compared to nanosecond laser processing, but the overall material removal rate was lower. Gao [56] proposed a chemically assisted ultraviolet femtosecond laser processing for diamonds; use of the chemically assisted treatment effectively removed the non-diamond carbon attached to the diamond surface during laser processing. The roughness of the unprocessed area and the laser processed area were reduced to 0.64 nm and 9.4 nm from 20.5 nm and 37.4 nm, respectively, and the prepared helical ribbon sheet structure device was used to obtain a good imaging focusing effect, a good image quality, and high quality of the surface. A good imaging focusing effect was obtained, and the effectiveness of the processing method was verified, as shown in Figure 8B. Compared with picosecond and nanosecond lasers with longer pulse times, femtosecond lasers are more suitable for the fine processing of special shapes for diamonds; however, the improvement in processing accuracy is at the expense of the processing rate.

## 4. Advances in the Main Applications of Laser Processing of Diamonds

The main industrialized application research of laser technology in diamond material processing mainly focuses on laser cutting, laser drilling, and micro-groove channel processing. At present, diamond CVD technology is becoming mature; how to refine the processing of diamonds has gradually become the main limiting factor in diamond applications; laser processing has gradually become the mainstream diamond processing method by virtue of its excellent processing performance.

### 4.1. Laser Cutting

The main ways to cut diamonds are water jet cutting, electric spark cutting and laser cutting. Laser cutting has unique advantages over other ways, i.e., contactless processing, high efficiency, small kerf, a small heat-affected zone, etc., which makes it an ideal method for processing diamonds. The current research on laser cutting focuses on finding narrow kerfs and large depths of cuts, etc. The smaller kerf taper minimizes the cutting loss. Park [57] proposed a processing method in an argon gas stream, which effectively avoids the unfavorable effects of material–plasma interactions, but the overall equipment construction is more complex and costly. To improve the depth of cut of CVD diamonds, Wang Ji et al. used a new type of acousto-optic modulated high-repetition frequency laser to process diamonds and investigated the effect of different process parameters on the processing effects (Figure 9A), in which the unidirectional depth of the cut could be up to 7.2 mm. With the increase in the depth of the diamond slit, the plasma-shielding effect during the cutting process was also enhanced gradually, which greatly limited the further increase in the depth of cut. Current research has not dealt with the cutting process for very large depths greater than 10 mm. The kerf taper is a parameter that measures the degree of variation in the depth and width of the channel, and, for laser cutting, a smaller kerf taper ensures maximum material utilization. Guo Qiang [58] conducted a systematic test on the cutting process of PCD composite sheets and studied the influence of process parameters such as laser power, cutting rate, pulse frequency, and refocusing on the cutting quality, and obtained excellent laser-cutting quality of PCD composite sheets with a slit width of 173.1 μm. The taper of the slit on one side of the slit was 5.90°, and the roughness was 0.65 μm. The high hardness and high thermal conductivity of diamonds make high demands on laser cutting. The formation of a highly collimated slit, the processing of ultra-thick diamond plates and the heat-affected zone, defects, etc. are the key problems to be solved in diamond laser cutting, while the development of short-pulse and ultrashort-pulse laser technology will significantly reduce the heat-affected zone and improve the cutting accuracy. Precise control of the laser beam and the development of new laser processing methods have become focuses for future development.

### 4.2. Laser Drilling

Currently, the main methods used for diamond microporous processing are laser etching and plasma etching. The latter has a better applicability for the simultaneous preparation of a large number of micropores, but the etching process is still immature, and it is difficult to complete the processing of deep holes. Therefore, diamond laser processing has become the first choice for rapid microporous molding. Martin [59] used a double-pulsed laser to ablate the diamond material on the silica sphere substrate for the preparation of diamonds. The double-pulse laser ablation of diamond material on a silica sphere substrate to prepare diamond micropores is shown in Figure 9B, in which funnel-shaped cones are cut across the diamond layer above, with depth-to-width ratios measured at the surface and at a 50% depth of 8:1 and 14.5:1, respectively; however, due to the thinness of the processed diamond film (65 μm), it is not very generalizable for practical processing. Natalie C. Golota [60] used a nanosecond laser (532 nm; 20 ns) to perforate a diamond, and the maximum depth-to-width ratio was 22:1 at 10,000 pulses. The maximum depth-to-width ratio was obtained by further increasing the cumulative number of pulses to above 40:1, as shown in Figure 9C.

### 4.3. Micro-Groove Processing

The diamond micro-groove structure can meet the urgent needs of the aerospace, electronics, and chemical industries in terms of heat-dissipation performance. Laser processing is currently the mainstream method of processing diamond micro-grooves. The key to the micro-groove structure is to cut and process the grooves with a high degree of collimation and to obtain a good surface quality. Shinoda [61] utilized the femtosecond laser to process 40 nm wide, 500 nm deep and 0.3 mm long periodic micro-grooves on the surface of a monocrystalline diamond with a depth-to-width ratio of 12:1. The 500 nm deep and 0.3 mm long periodic microgrooves on the single crystal diamond surface with a depth-to-width ratio of 12:1 and an average groove pitch of (146 ± 7) nm between the microgrooves are shown in Figure 10A. Qi Zhina [38] is committed to the study of diamond microchannel heat-dissipation preparation and heat-sink performance. In one study, high-energy laser-beam flow ablation was used to prepare ultra-thick diamond microchannels (Figure 10B), based on a single-phase heat transfer system. The thermal conductivity of the diamond microchannel heat sinks ranged from 5637.1 to 11,447.2 W/(m^2^K), which is 37% to 73% higher than that of aluminum microchannel heat sinks of the same shape. This translates to a 40% reduction in the flow requirement for diamond heat sinks when translated into the volume flow rate of the fluid in the heat sink. Dudek [62] processed polycrystalline diamond samples with different microstructures using a nanosecond laser (355 nm) and obtained microstructures with precise geometries (Figure 10C), good perpendicularity, and deep channels, thereby reconfirming the feasibility of lasers for the fabrication of diamond microfluidic devices. A numerical analysis of liquid–solid conjugate heat transfer in 3D diamond microgrooves with different cross-sectional shapes showed that rectangular microgrooves have better heat transfer performance than triangular and trapezoidal microgrooves. Ensuring the consistency of the overall microgroove channel depth is a major point of concern in the processing of large-area microgrooves. Yasuhiro Okamoto [39] proposed a two-step scanning method that is practical for the preparation of large areas on the surface and provides better control of the microgroove depth. This method can greatly reduce the machined surface roughness of single-crystal diamonds and can be used as a high-quality single-crystal diamond micro-shape machining method.

## 5. Conclusions

C is a group of elements widely found in nature, and carbon atoms form various allotropes by their different arrangements. Diamond crystals are atomic crystals that are also composed of carbon. The carbon atoms in the crystal lattice are covalently bonded to each other in the form of sp3 hybridization, which provides very high thermal conductivity, hardness, sonic velocity, and carrier mobility at room temperature. In addition, it has unique irradiation stability, chemical inertness, biocompatibility, and many other important properties that define the many applications of diamond materials.

(1) Picosecond and femtosecond lasers process diamonds with high precision and quality, but with low processing efficiency. Millisecond and nanosecond lasers have longer pulse durations, which inevitably leads to the formation of heat-affected zones and the production of defects during processing, and cannot meet industry requirements and the demand for high-quality diamond processing. It is necessary to optimize the laser process parameters to ensure a high material removal rate during the process and at the same time to ensure a high-quality processed surface, and to further reduce the generation of heat-affected zones in the processed samples;

(2) The laser processing of diamonds is always accompanied by thermal stress and deformation problems, especially for self-supported, large-size diamond films. The accumulation of heat during processing and the release of stress during material removal inevitably increase the deformation of diamond wafers, which create difficulties in subsequent applications and processing;

(3) For high-quality, optical- and electronic-grade diamonds, laser processing will introduce defects to the surfaces or sub-surface layers of diamonds, deteriorating their optical and electrical properties. The problem of damage from laser processing is currently one of the key points limiting expansion in the field of diamond laser processing.

Although there are still many problems with the laser processing of diamonds, laser processing will remain one of the main technologies in the field of diamond processing in the future. Laser-processing technology will also become more mature to meet various processing needs, and gradually develop higher efficiency and precision, lower damage rates, higher integration, and production automation. The application prospects of diamond laser processing will continue to broaden for the foreseeable future.

## Figures and Tables

**Figure 2 micromachines-15-00547-f002:**
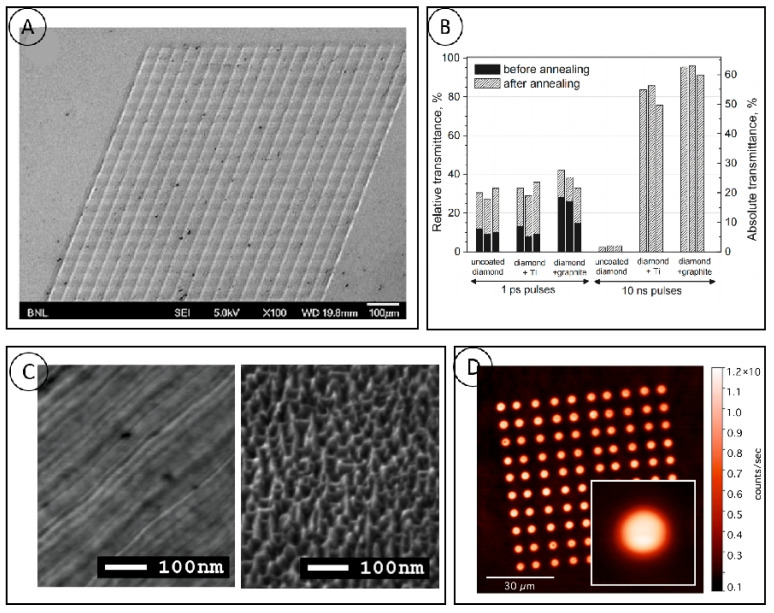
(**A**) SEM images of laser ablation [27]; (**B**) absolute and relative transmittance of the pockets fabricated by 1 ps and 10 ns pulses with different absorbing coatings [29]; (**C**) scanning electron microscope images of diamond surface before and after laser nanoablation [30]; and (**D**) fluorescence image of NV color center array [31].

**Figure 3 micromachines-15-00547-f003:**
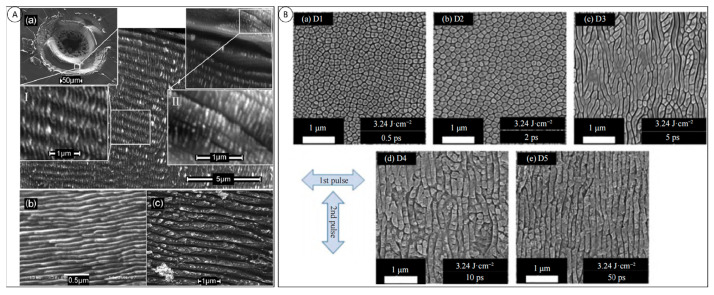
(**A**) Scanning electron microscope images of diamond surface irradiated by 800 nm femtosecond laser. (a) SEM image of graphite surface irradiated by 532 nm ps laser. (b) SEM image of graphite surface irradiated by 400 nm fs laser. (c) SEM image of graphite surface irradiated by 1300 nm ps [34]; and (**B**) with the increase in pulse time delay, the evolution of diamond surface morphology array [35].

**Figure 4 micromachines-15-00547-f004:**
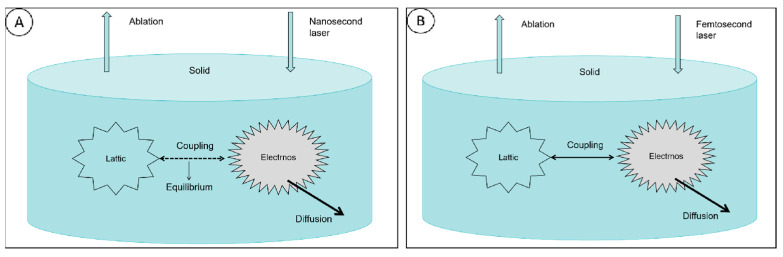
(**A**,**B**) Nanosecond laser and femtosecond laser have different interaction models with electron and lattice, respectively.

**Figure 5 micromachines-15-00547-f005:**
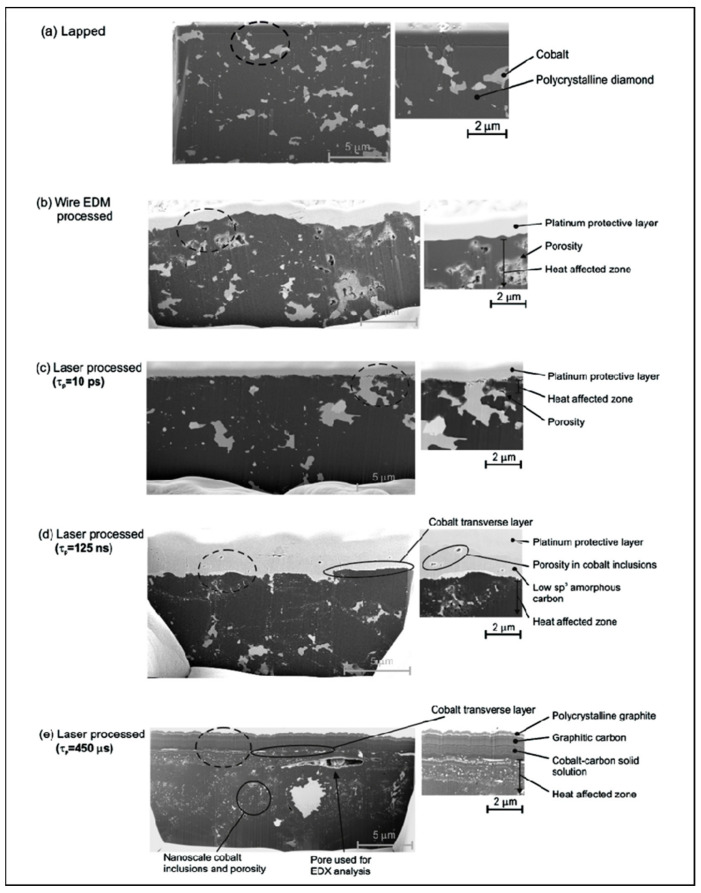
The cross-section of PCD composite was obtained by microsecond laser processing: (**a**) lapping; (**b**) wire EDM; and (**c**–**e**) laser [40].

**Figure 6 micromachines-15-00547-f006:**
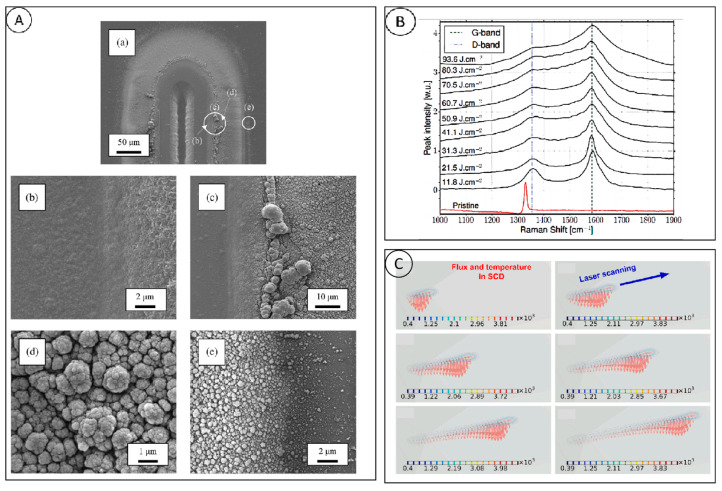
(**A**) SEM image of nanosecond laser processing: (a) the entire groove, (b) thin layer of debris deposition, (c) large debris particles, (d) small round particles, (e) decrease in particle size [42]; (**B**) Raman spectral images of machined pits under different laser fluxes [43]; and (**C**) heat conduction and temperature distribution in the single crystal diamond at different scanning time, in which the arrowheads indicate the conductive heat flux direction [44].

**Figure 7 micromachines-15-00547-f007:**
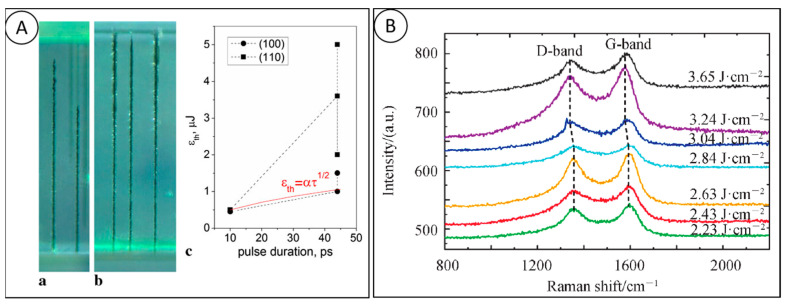
(**A**) Schematic diagram of nanosecond laser processing and optical picture of graphitized microstructure device. (a) longitudinal scanning processing, (b) horizontal scanning processing, (c) thresholds of bulk microstructure formation for two pulse durations and two beam directions [46]; and (**B**) Raman at the center position of diamond microgroove by picosecond laser ablation at different laser energies spectrum [49].

**Figure 8 micromachines-15-00547-f008:**
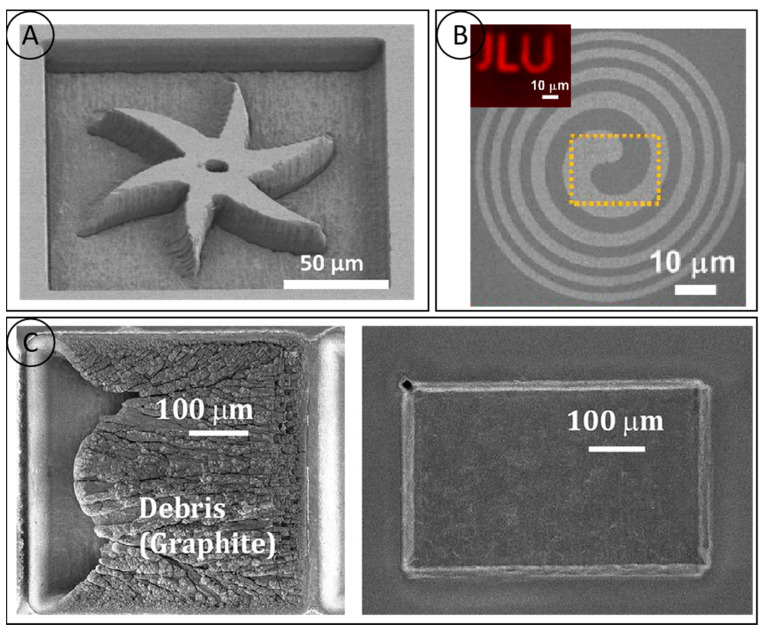
(**A**) SEM image of the surface of single crystal diamond processed by femtosecond laser, bending structure processed by laser pulse (energy is 1.2 mJ) [54]; (**B**) SEM image of diamond spiral strip structure processed by fs laser, the insertion diagram is the imaging effect; and (**C**) surface contrast topography of diamond processed by nanosecond laser and femtosecond laser [55].

**Figure 9 micromachines-15-00547-f009:**
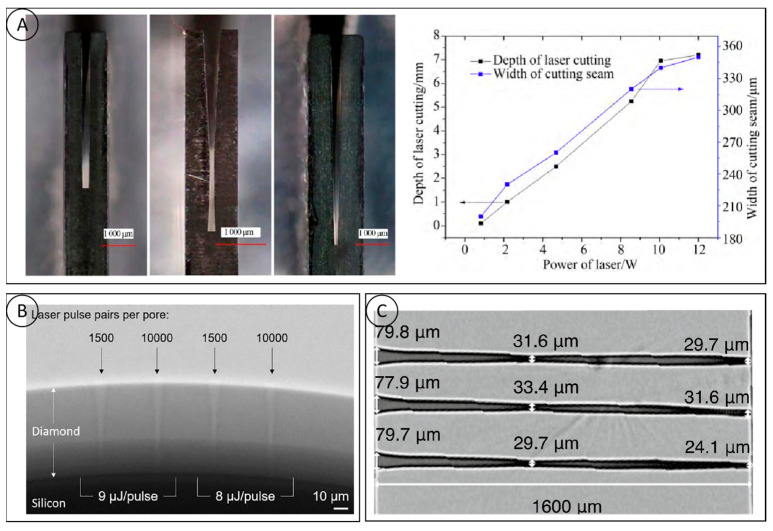
(**A**) The cutting effect diagram of different focus settings and the relationship between laser power and cutting depth and slit width [57]; (**B**) radiograph of micropores fabricated in a diamond on silicon sphere substrate by the double-pulse laser ablation method [59]; and (**C**) the average depth to width ratio obtained in diamond is 40∶1 [60].

**Figure 10 micromachines-15-00547-f010:**
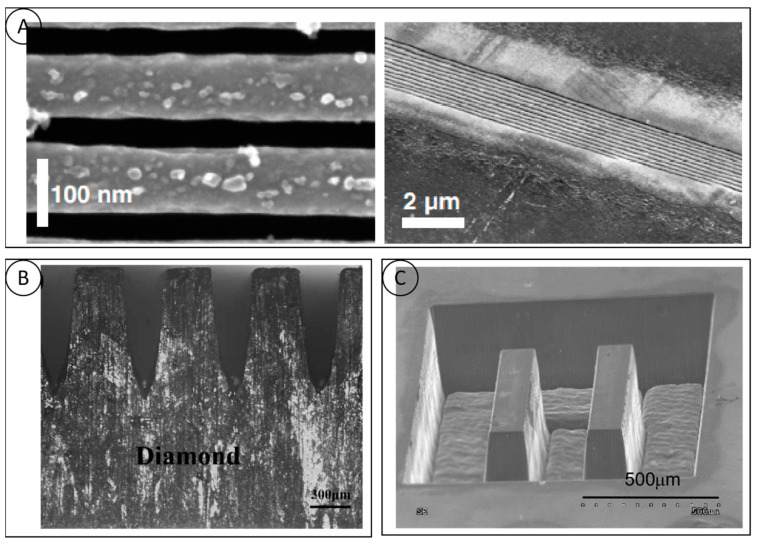
(**A**) SEM image of grooves obtained at a pulse energy of 72 nJ [61]; (**B**) side view of the diamond microchannel [38]; and (**C**) SEM image of microgroove prepared by nanosecond laser [62].

**Table 1 micromachines-15-00547-t001:** Classification and properties of diamond.

Type	Mass Fraction in Nature	Crystalline Type	Impurity	Color	Electric Resistance /(Ω·cm)
Ⅰa	~98%	Polycrystalline	Mass fraction of N is about 2 × 10⁻³ N is in aggregation state	Colorless to yellow	10⁴–10¹⁶ (insulator)
Ⅰb	~0.1%	Small single crystal	Mass fraction of N and metal inclusion are 10⁻⁴–10⁻¹ and 10⁻²–10⁻¹, respectively N is in dispersion state	Green to Brown	10⁴–10¹⁶ (insulator)
		Large single crystal	Mass fraction of N is 10⁻⁶–10⁻⁴ N is in dispersion state	Yellow	10⁴–10¹⁶ (insulator)
IIa	1–2%	Single crystal	Mass fraction of N is 10⁻⁶	Colorless	10¹⁶ (insulator)
IIb	~0	Single crystal	Mass fraction of N and B are 10⁻⁶ and 10⁻⁴, respectively	Blue	10¹–10⁴ (P type semiconductor)

**Table 2 micromachines-15-00547-t002:** Types of lasers used in diamond processing.

Type	Nd:YAG		Cu	$CO_{2}$	$Ar^{+}$	KrF	ArF	$\mathrm{Ti:}Al_{2}O_{3}$
Wavelength/nm	1.064	532	510.5	10,600	488	248	193	800
Energy/eV	1.17	2.33	2.42	0.12	2.54	5.0	6.42	1.55
Mode	Pulse/continuous		Pulse	Pulse	Continuous	Pulse	Pulse	Pulse

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
