# Peer review of "Research and Application Progress of Laser-Processing Technology in Diamond Micro-Fabrication"

_micromachines, 2024, doi:10.3390/mi15040547_

Round 1
Reviewer 1 Report
Comments and Suggestions for Authors
The paper explains the fundamental principle of diamond laser processing, introduces the interaction mechanism between various types of lasers and diamond materials, focuses on analyzing the current development status of various modes of laser processing of diamond recently, briefly discusses the relevant applications in diamond cutting, micro-hole forming, and micro-groove machining, etc., and finally discusses the issues, challenges , and potential future advancements of laser technology in the field of diamond processing at this point. However, the paper needs to be revised as follows before publication: 1. In section 1, Page 2, the author mentioned in Table 1 that the color of Type Ⅰb is “Green to”, and the author is invited to explain. At the same time, ask the author to check whether the title of the form is appropriate. 2. In section 1, Page 2, the description of Table 2 is missing in the article. 3. In section 2.3, Page 8, the author lacks an explanation of variable p in formula 1, please explain its physical meaning. At the same time, the font for physical quantities should be in italics, please modify it. 4. In the image quoted in the article, the author makes a new annotation of the image, which is in conflict with the annotation of the original image, and the author should check whether the description of the image is appropriate. 5. In the article, the format of the first part and the last part of the text is not uniform, please the author to revise the text format of the article.
Comments on the Quality of English Languagecheck
Author Response
- In section 1, Page 2, the author mentioned in Table 1 that the color of Type Ⅰb is “Green to”, and the author is invited to explain. At the same time, ask the author to check whether the title of the form is appropriate.
Answers: Thanks. In Table 1, the color of Type I b is "Green to Brown".
- In section 1, Page 2, the description of Table 2 is missing in the article.
Answers: Thanks. I have added the description of Table 2.
Lines 271 to 282 :
The main types of lasers used in current research on laser processing of diamond materials are shown in Table 2. Most of them have pulse frequencies between 0.1 Hz and 100 kHz, and the shortest pulse widths are on the femtosecond scale. By choosing the appropriate laser wavelength, pulse width, and power parameters, diamond can be processed with high-quality and special shapes, and its processing accuracy can reach the micrometer or nanometer level.
- In section 2.3, Page 8, the author lacks an explanation of variable p in formula 1, please explain its physical meaning. At the same time, the font for physical quantities should be in italics, please modify it.
Answers: Thanks. I added the explanation for the variable ρ in Equation 1, ρ is the density of the material. I have also changed the font of the physical quantities to italics.
- In the image quoted in the article, the author makes a new annotation of the image, which is in conflict with the annotation of the original image, and the author should check whether the description of the image is appropriate.
Answers: Thanks. I have made new annotations to the pictures and modified the annotations that conflict with the original pictures
- In the article, the format of the first part and the last part of the text is not uniform, please the author to revise the text format of the article.
Answers: Thanks for the overall comments. I have modified it.

Reviewer 2 Report
Comments and Suggestions for Authors
The manuscript “Research and application progress of laser processing technology in diamond micro-fabrication” lists different processing techniques to produce diamond, focusing on one of them: laser processing. The manuscript shows the fundamentals and the effect of parameters variation on the diamond properties. Before the acceptance for publishing in Micromachines journal, the reviewer lists some comments that the authors must consider to improve the manuscript’s quality.
Comments:
The reviewer recommends using simply N instead of “Nitrogen” and C instead of “carbon” in the whole manuscript.
Generally, the images have low resolution, and better images must replace the actual ones.
The reviewer recommends reviewing all the units written in the manuscript and adopting a standard. Some examples of faults are listed below, but the authors must to review the whole manuscript.
Line 24: “0.155” and in Line 25 “nm”. 0.155 and nm must remain in the same line, not breaking it. The reviewer recommends reviewing the whole manuscript, searching for this type of fault.
Line 27: “1.2×10-6 K-1”. The reviewer recommends reviewing the whole the manuscript, searching for this type of fault. The correct is 1.2×10–6 K–1.
Line 40: “The HPHT method and the CVD method are used to make gem-quality diamonds.” The reviewer recommends: “High pressure high temperature (HPHT) and chemical vapor deposition (CVD) methods are used to make gem-quality diamonds.”
Line 64: typo: “Table 1. Table 1”.
Line 65: “10⁴⁻10¹⁶”. The reviewer recommends reviewing the whole manuscript, searching for this type of fault. The correct is 104 ~ 1016.
Line 161: “(a) SEM images of laser ablation[27].” The reviewer recommends insert a space before the reference citation. The correct is “a) SEM images of laser ablation [27].” The reviewer recommends reviewing the whole manuscript, searching for this type of fault.
Line 168: “John Smedley et al. [27]…”, and Line 175: “…by T.V. Kononenko et al. [28].” The reviewer recommends using only the authors’ surname. The correct are “Smedley et al. [27]…” and “by Kononenko et al. [28].” The reviewer recommends reviewing the whole manuscript, searching for this type of fault.
Line 195: “A representative work is Chen et al. from the University of Oxford [29], …” The reviewer does not see any importance in presenting the authors’ affiliation. The reviewer recommends: “A representative work was written by Chen et al. [29], who…”
Line 198: “… diamond in 2017, …” The reviewer does not see relevance in list the publishing year and recommends delete this info from the manuscript.
Line 204: typo: “27n J”.
Line 205 and 207: typo: “19n J”.
Line 211: The reviewer recommends “Chen et al. [30] reported…” instead to “Further, in 2019, the same group [30] went on to report…”.
Line 233: “1.9 J/cm2”. The correct is “1.9 J·cm–2”. The reviewer recommends reviewing the whole manuscript searching for this type of fault.
Line 357-361: The image nomenclature (a), (b), (c) … is confused. The reviewer recommends editing the image and its title for a better comprehension.
Line 408: The reviewer did not find this image (Fig. 8a) in the reference cited by the authors. Please provide a correct reference for the image.
Line 538: The reviewer did not find this image (Fig. 10a) in the reference cited by the authors. Please provide a correct reference for the image.
Line 538: The reviewer did not find this image (Fig. 10c) in the reference cited by the authors. Please provide a correct reference for the image.
Author Response
Response to 1-7 : Thanks. We have revised the article following the comments.
Response to 8 and 11 : Thanks. We have deleted this part of the article and added other content.
Lines 207 to 213 :Recently, researchers have conducted a number of studies around the use of femtosecond lasers to induce the generation of color centers. V. Kononenko presented the concept of color centers prepared by surface nano-ablation [29,30], which were efficiently and controllably generated by a femtosecond laser acting on the diamond surface, and also explored the influence of the laser parameters on the formation of NV color centers during the laser-diamond interaction, as shown in Fig. 2C-D.
Response to 9 : Thanks. It has been deleted as you suggested.
Response to 12: Thanks. We have revised the article following the comments.
Response to 13: Thanks. We have modified the picture and corresponding description according to your suggestion.

Round 2
Reviewer 2 Report
Comments and Suggestions for Authors
The manuscript “Research and application progress of laser processing technology in diamond micro-fabrication” lists different processing techniques to produce diamond, focusing on one of them: laser processing. The authors improved the manuscript’s quality by accepting some reviewer’s recommendations; however, some minor corrections still remain before its acceptance for publishing in Micromachines journal.
Comments:
In the first round, the reviewer recommended: “using simply N instead of ‘Nitrogen’ and C instead of ‘carbon’ in the whole manuscript”. However, in Table 1, it was not considered, although the authors replied to this comment in the first round: “Thanks for the overall comments. I have modified it.”. The authors should reply why they did not accept the reviewer’s suggestion or effectively modify the manuscript. Please review the whole manuscript, as requested previously.
The reviewer recommended in the firs review round using only the authors’ surname for citations in the manuscript. However, the manuscript still has citations in different formats, e.g., “M.S. Komlenok [24]” (Line 137), “Smedley et 176 al. [27]” (Line 176), and “Magdalena Forster [33]” (Line 214). The manuscript has to follow a standard for presenting the citations, following the journal format. The authors must review the whole manuscript, searching for this type of fault.
Figure 5 does not need the subtitle “A”, because it is one only image, and the small images that compose the Figure are already named as a, b, c, d, and e.
In Figure 10 B is indicated (c) too, probably from the original document references. The reviewer recommends editing this image to mask this (c), avoiding reading confusion. The reviewer does the same consideration for Figure 8B, which has (a1) name too; Figure 1A, which has a, b, and c names too; Figure 6A, which has a, b, c, d, and e names too; and Figure 2C, which has the name a too.
Author Response
Thanks for the reviewer's time, please see the attachment.
